# Patient satisfaction and patient accessibility in a small fiber neuropathy diagnostic service in the Netherlands: A single-center, prospective, survey-based cohort study

**Margot Geerts**[1]*, **Janneke G. J. Hoeijmakers**[1○], **Brigitte A. B. Essers**[2○], **Ingemar S. J. Merkies**[1,3○], **Catharina G. Faber**[1○], **Mariëlle E. J. B. Goossens**[4○]

1 Department of Neurology, School of Mental Health and Neuroscience, Maastricht University Medical Center+, Maastricht, The Netherlands, 2 Department of Clinical Epidemiology and Medical Technology Assessment, Maastricht University Medical Center+, Maastricht, The Netherlands, 3 Department of Neurology, Curacao Medical Center, Willemstad, Curacao, 4 Department of Rehabilitation Research & Department of Clinical Psychological Sciences, Maastricht University, Maastricht, The Netherlands

○ These authors contributed equally to this work.
* m.geerts@mumc.nl

## Abstract

### Introduction

Small fiber neuropathy (SFN) is a common cause of neuropathic pain in peripheral neuropathies. Good accessibility of diagnostics and treatment is necessary for an accurate diagnosis and treatment of SFN. Evidence is lacking on the quality performance of the diagnostic SFN service in the Netherlands. Our aim was to determine the patient satisfaction and -accessibility of the diagnostic SFN service, and to identify areas for improvement.

### Methods

In a single-center, prospective, survey-based cohort study, 100 visiting patients were asked to fill in the SFN patient satisfaction questionnaire (SFN-PSQ), with 10 domains and 51 items. Cut-off point for improvement was defined as ≥ 25% dissatisfaction on an item. A chi-square test and linear regression analyses was used for significant differences and associations of patient satisfaction.

### Results

From November 2020 to May 2021, 98 patients with SFN-related complaints filled in the online SFN-PSQ within 20 minutes. In 84% of the patients SFN was confirmed, average age was 55.1 (52.5–57.8) years and 67% was female. High satisfaction was seen in the domains 'Waiting List Period', Chest X-ray', 'Consultation with the Doctor or Nurse Practitioner (NP)', 'Separate Consultation with the Doctor or NP about Psychological Symptoms', and 'General' of the SFN service. Overall average patient satisfaction score was 8.7 (IQR 8–10) on a 1-to-10 rating scale. Main area for improvement was shortening the 8-week period for

**Data Availability Statement:** All relevant data are within the manuscript and its Supporting Information files.

**Funding:** The author(s) received no specific funding for this work.

**Competing interests:** I have read the journal's policy and the authors of this manuscript have the following competing interests: Janneke G.J. Hoeijmakers reports a grant from the Prinses Beatrix Spierfonds (W.OK17-09), outside the submitted work. Catharina G. Faber reports grants from European Union's Horizon 2020 research and innovation program Marie Sklodowska-Curie grant for PAIN-Net, Molecule-to-man pain network (grant no. 721841), grants from Prinses Beatrix Spierfonds, grants from Grifols and Lamepro for a trial on IVIg in small fiber neuropathy, other from Steering committees/advisory board for studies in small fiber neuropathy of Biogen/Convergence and Vertex, outside the submitted work. Ingemar S.J. Merkies reports grants and non-financial support from Grifols, grants from Lamepro, during the conduct of the study; other from Participation in steering committees of the Talecris ICE Study, CSL Behring, LFB, Novartis, Octapharma, Biotest and UCB, outside the submitted work.

receiving the results of the diagnostic testing ($p < 0.05$). General health status was statistically significant associated with patient satisfaction ($p < 0.05$)

## Conclusion

A good reflection of the high patient satisfaction and -accessibility of the SFN-service is shown, with important points for improvement. These results could help hospitals widely to optimize the logistic and diagnostic pathway of SFN analysis, benchmarking patient satisfaction results among the hospitals, and to improve the quality of care of comparable SFN services.

## Introduction

Small fiber neuropathy (SFN) is a common peripheral chronic pain condition, caused by impaired small thin myelinated and unmyelinated nerve fibers, leading to severe neuropathic pain and autonomic dysfunction [1, 2]. The minimal incidence and prevalence rates of SFN are 13.3 to 52.9 cases per 100,000 inhabitants [3, 4]. Although various etiological conditions, like diabetes mellitus, autoimmune diseases, and sodium channel gene mutations are associated with SFN, in 53% no identifiable cause can be found [5]. Current treatment of SFN is based on treating the underlying cause and symptomatic pharmacological treatment with antidepressants and anticonvulsants, but suboptimal pain relief and side effects frequently occur [6–8]. Quality of life (Qol) is severely affected by SFN compared to healthy subjects [9, 10], and in more than one-third of the patients with SFN anxiety and/or depressive symptoms are found [11].

Early diagnosis and improving patient satisfaction and -accessibility to SFN-health care are the basic aims for patients with SFN-related complaints. An SFN diagnosis could potentially help these patients find a possible underlying cause, and could lead to specific therapy for this cause, neuropathic pain treatment, contact with support groups and self-management [12]. However, the diagnostic and treatment pain services of European specialist pain clinics varies widely [13]. Therefore, to improve the quality of SFN care, a more targeted diagnostic SFN service has been adopted and fine-tuned through the years in the Netherlands based on patients' input. A detailed report of the diagnostic SFN service has been published earlier [14], and a brief overview is given below. The diagnostic SFN service includes a visit at the outpatient clinic (OC) or an 1-day stay at the neurological day care unit (NDCU) for diagnostic tests, consultation and neurological examination. Neurological analysis includes nerve conduction studies (NCS), which is an electro-diagnostic test for classifying a possible neuropathy [15], and quantative sensory testing (QST) for quantifying the thresholds of sensory perception by large and small nerve fibers [16]. Finally, a multidisciplinary team with all potential stakeholders ((neurologists, pain specialists, psychiatrists, geneticists, nurse practitioners (NPs), nurses, physiatrists)) discusses the diagnostic results in order to select evidence-based treatment options, tailored to the patient's personal situation.

In acting on patient-centeredness in the clinical setting, more patient involvement is needed so that providers pay attention to patients' identities, concerns and preferences [17]. Patient experiences are an important source of information for measuring quality of care from a patient perspective [18]. Important criteria of variables to promote patient-centered care in the clinical consultation are consultation processes, patient satisfaction, patient healthcare behaviors and health status [17]. These variables were included in this research by using the SFN

patient satisfaction questionnaire (SFN-PSQ), which was developed in an earlier qualitative study to measure the quality performance of the diagnostic SFN service from the patients' perspective. Through item generation, expert opinions, and interviews by patients the content validity and feasibility of the SFN-PSQ was developed and validated. [*ref Geerts et al. 2023. Development, validation and feasibility of a patient satisfaction questionnaire for evaluating the quality performance of a diagnostic small fiber neuropathy service*: *A qualitative study*, *submitted at Health Expectations on October 20*[th], *2023*]. The goal of the current study was to measure patient satisfaction, and -accessibility of the diagnostic SFN service by using the SFN-PSQ in an online survey. These findings can be used for benchmarking patient satisfaction results among the hospitals with comparable diagnostic SFN services, so that quality improvements can be implemented on important topics of SFN care.

## Methods

### Study design

The study consisted of a single-center, prospective, survey-based cohort study to determine the patient satisfaction and -accessibility of a diagnostic SFN service in the Netherlands.

The cohort study was carried out at the SFN Center at the Maastricht University Medical Center+ (Maastricht UMC+), in the Netherlands. This SFN Center is the only specialized SFN center in the Netherlands working according to the latest evidenced-based diagnostic standards [2, 5, 16, 19]. On a yearly basis, the SFN Center has a limited access for diagnosing 500 patients with SFN-related complaints.

Patients fill in a set of validated questionnaires before visiting SFN Center, which are described in the next section.

There are some differences between the logistic and personnel features of the OC and the NDCU. The NDCU has more facilities in the waiting room and a staff member is present to take patients to the diagnostic tests. In the OC, the patient consultation is done by a medical resident in the role of medical doctor instead of NPs or medical researchers at the NDCU.

### Study procedure and data collection

From November 12th 2020 to May 4th 2021, all consecutive patients of 18 years and older, who completed the neurological analysis of the diagnostic SFN service at the OC or NDCU, were asked, at the final OC appointment for discussing the results with their SFN caregiver, to participate in the online survey with the SFN-PSQ.

Age, sex, education level, anxiety and depression scores and health status can significantly affect patient satisfaction [20–23], therefore these baseline patient characteristics like were collected.

The completed level of education of patients was assessed in three categories: (1) low education level ('elementary school', 'practical education' or 'lower secondary vocational education'); (2) medium education level ('pre-vocational secondary education', 'secondary vocational education', 'senior general secondary education' or 'pre university education'); and (3) high education level ('university of applied sciences' or 'university') [24].

The Hospital Anxiety and Depression Scale (HADS) questionnaire was used for assessing anxiety and depression scores. The HADS-questionnaire has two subscales, for measuring anxiety (HADS-A) and depression (HADS-D). Both subscales consist of seven questions, with a range from 0 to 21. Higher scores indicating more symptoms of anxiety and depression [25]. A patient consultation at the Psychiatry Department can be scheduled if the HADS score is $\geq$ 14.

General health status was completed on a 5-point Likert-type response category (poor, fair, good, very good, or excellent), which was part of a validated and feasible core questionnaire for the assessment of patient satisfaction (COPS) [26].

Digital data collection was realized at home with an internet-based electronic environment (SelectSurvey-net, a digital questionnaire system according to Good Clinical Practice guidelines) in collaboration with the Clinical Trial Center Maastricht, the Netherlands. Patients were given access by a hyperlink to an internet-based electronic environment to complete the SFN-PSQ online. If patients were not able to complete the SFN-PSQ online, a paper version by post was provided. Incapacitated patients and patients who do not speak or understand Dutch were excluded from participation.

### Small fiber neuropathy patient satisfaction questionnaire (SFN-PSQ)

A complete report of the new developed small fiber neuropathy patient satisfaction questionnaire (SFN-PSQ) has been described elsewhere [*Geerts et al. 2023*]. The SFN-PSQ is a valid and clinically useful questionnaire for measuring the quality performance of the diagnostic SFN service from the patients' perspective. The SFN-PSQ consists of ten domains: (A) General Health Status, with one item; (B) Waiting list period, with six items; (C) Reception and Stay in the waiting room, with nine items; (D) Chest x-ray, with four items; (E) Nerve Conduction Studies, with four items; (F) Treatment by the medical doctor (MD) or nurse practitioner (NP), with eleven items; (G) Consultation by the Hospital Psychiatrist (yes/no); (H) Consultation by the Hospital Psychiatrist, with six items; (I) Follow-up Services, with seven items; (J) Overall Quality Assessment, with 3 items and in total of 51 questions, with 48 Likert-type items ('not at all satisfied', 'a little bit satisfied', 'mostly satisfied', 'yes, completely satisfied', 'not applicable'), one question with a 5-point Likert-type items (1: bad to 5: excellent), two multiple-choice questions with two options and a question with a 1-to-10 rating scale (S1 Appendix).

### Data analysis

Descriptive statistics were used to describe the sociodemographic (e.g. age, sex, education), clinical (diagnosis of SFN, anxiety, depression) and logistic features (type of consultation) as well as the responses and missing data on question level the SFN-PSQ. Patients who did not fill in 25% or more of the SFN-PSQ were excluded from analysis [27, 28]. All patients with less than 25% missing scores were labeled as missing data. We defined a cut-off point for improvement at $\geq$ 25% dissatisfaction (no, not at all satisfied or a little satisfied) on an item. Differences in patient satisfaction and demographics, diagnosis SFN, and type of consultation were tested with a Chi-square test, appropriate for the categorical variables. Differences in patient satisfaction and clinical features like age, anxiety and depression were tested with a Kruskal-Wallis test, suitable for continuous variables. Multivariate linear regression analyses were performed to investigate the association of age, sex, education level, anxiety and depression scores and health status, with patient satisfaction. These variables were selected based on their relevance according to the literature [20–23]. A backward stepwise method was used to test for interaction between the independent variable on all outcomes using an $\alpha$ of 0.05. All analyses were performed using IBM Statistics SPSS version 25.0.

### Ethics

The study protocol was not subject to the Medical Research Involving Human Subjects Act (WMO), which was confirmed by the Medical Ethics Committee of the Maastricht UMC+ (2019–1049). Informed consent was digital written obtained by a hyperlink to an internet-

based electronic environment to complete the SFN-PSQ online before participating in the study, according to the principles of the Declaration of Helsinki [29].

# Results

## A. Patient characteristics and general health status

A total of 103 visiting patients were asked to participate in the study, of which 98 (95%) completed the SFN-PSQ. Three patients gave no permission to participate and two patients did not fill in the SFN-PSQ. Table 1 presents feasibility of the SFN-PSQ, logistic features and demographics of the overall study population (n = 98), the patients with SFN (n = 82), and patients

**Table 1. Logistic features and demographics of the study population with SFN and without SFN.**

| | Overall | SFN | No SFN | P |
|---|---|---|---|---|
| | (N = 98) | (n = 82) | (n = 16) | |
| **Duration of completing the questionnaire (minutes), average (CI)** | 13.8 (11.9–15.7) | 15.7 (10.2–21.2) | 13.4 (11.4–15.4) | 0.799 |
| **Age (years)** | | | | 0.348 |
| average (CI) | 55.1 (52.5–57.8) | 54.7 (51.8–57.5) | 57.4 (49.8–65.0) | |
| median (IQR) | 57.0 (46.8–65.0) | 57.0 (46.0–63.5) | 60.0 (47.8–65.8) | |
| **Sex, (n, %)** | | | | 0.060 |
| Female | 66 (67.3) | 52 (63.4) | 14 (87.5) | |
| **Level of education, n (%)** | | | | 0.333 |
| Low | 15 (15.3) | 12 (14.6) | 3 (18.8) | |
| Medium | 53 (54.1) | 47 (57.3) | 6 (37.5) | |
| High | 30 (30.6) | 23 (28.0) | 7 (43.8) | |
| **Type of consultation (n,%)** | | | | **0.004** |
| NDU | 80 (81.6) | 71 (86.6) | 9 (56.3) | |
| OU | 18 (18.4) | 11 (13.4) | 7 (43.8) | |
| **Diagnostic tests of SFN, n (%)** | | | | **< 0.001** |
| Normal IENFD and normal QST | 16 (16.3) | 16 (16.3) | - | |
| Abnormal IENFD and abnormal QST | 36 (36.7) | 36 (36.7) | - | |
| Abnormal IENFD | 15 (15.3) | 15 (15.3) | - | |
| Abnormal QST | 31 (31.6) | 31 (31.6) | - | |
| **Visiting alone or with an attendant (n,%)** | | | | 0.684 |
| With an attendant | 63 (64.3) | 52 (63.4) | 11 (68.8) | |
| **One way travel distance (kilometers)** | | | | **0.006** |
| average (CI) | 193.2 (177.7–208.7) | 194.6 (177.7–211.5) | 186.2 (142.0–230.4) | |
| median (IQR) | 203.5 (131.0–241.5) | 202.5 (138.5–241.5) | 210.0 (106.3–242.3) | |
| **HADS, median (IQR)** | | | | |
| Anxiety | 7.0 (3.3–10.0) | 7.0 (3.5–10.0) | 7.1 (3.0–11.0) | **0.030** |
| Depression | 5.0 (3.0–9.0) | 6.4 (3.0–9.0) | 5.5 (3.0–9.0) | 0.356 |
| Missings | 18 (18.4) | 13 (15.9) | 5 (31.1) | |
| **1. General health status (n,%)** | | | | 0.450 |
| Bad | 11 (11.2) | 9 (11.0) | 2 (12.5) | |
| Moderate | 57 (58.2) | 50 (61.0) | 7 (43.8) | |
| Good | 28 (28.6) | 21 (25.6) | 7 (43.8) | |
| Very good | 2 (2.0) | 2 (2.4) | 0 (0.0) | |

NDCU: Neurological Day Care Unit. OC: Outpatient Clinic. CI: Confidence Interval. IQR: Inter Quartile Range. SFN: Small Fiber Neuropathy. IENFD: Intra-epidermal Nerve Fiber Density. QST: Quantitative Sensory Testing. HADS: Hospital Anxiety and Depression Scale.

with no SFN (n = 16). The average duration for completing the SFN-PSQ was approximately 14 minutes (CI 11.9–15.7), and more than 80% of the patients completed the SFN-PSQ within 20 minutes. The median age of the SFN-population was 57.0 (range 46.0–63.5) years and 63% was female. The majority of the patients with SFN had a medium or high educational level (86%). Almost 87% of the patients with SFN were analyzed at the one day stay at the NDCU (p = 0.004). In 37% of the patients the diagnosis of SFN was confirmed by an abnormal IENFD and abnormal QST (p < 0.001). Patients with SFN had a significant higher one way travel distance, with an average of 195 kilometers (CI 177.7–211.5, p = 0.006). The median scores of the SFN-population for symptoms of anxiety and depression were 7.0 (IQR 3.5–10.0, p = 0.006) and 6.4 (IQR 3.0–9.0) respectively. The general health status was rated as bad or moderate in 72% of patients with SFN.

## B. Waiting List Period

Almost 85% of all patients was largely to completely satisfied about the acknowledgment with expected time until the date of neurological analysis, the information about the neurological analysis and the internet link for information about SFN, see Table 2. More than 50% of all patients had to make arrangements at home or at work to be able to come to the hospital, mostly transport (34.4%) and work leave (28.1%). Almost 70% of all patients was not or a little satisfied about the waiting time until the neurological analysis; the average waiting time was 10.5 (SD 5.6) months.

## C. Reception area and time in the waiting area

A majority of all patients was largely to completely satisfied about with the accessibility, privacy at the front desk, and the assistance of the receptionist (Table 3). Furthermore, more than 75% of all patients was largely to completely satisfied with the welcome, information and guidance of the staff member who attended them throughout the day at the NDCU. A majority was largely to completely satisfied about the skills of the staff member at drawing blood and only 2% of all patients experienced the blood draw as inconvenient or unpleasant. Nearly 90% of all patients was satisfied about the facilities in the waiting room.

## D—E. Chest X-ray and nerve tests (NCS and QST)

Overall, all patients were largely to completely satisfied about the item concerning the welcome, information and guidance, personal attention, and expertise from the staff members of the chest x-ray and nerve tests (80.6% - 85.7% and 92.9%– 94.1% respectively, Table 4).

## F. Consultation with the Doctor/NP

In Table 5 the results of patient satisfaction of the consultation by the Doctor of NP are presented and overall, patients were largely to completely satisfied with personal attention (98.0%), expertise (96.1%) and privacy (100%). Furthermore, patients were largely to completely satisfied about asking about their own ideas, expectations or experiences, the impact of their symptoms on their daily life, and their lifestyle (including smoking, exercise, diet) (all ≥ 95.0%). Nearly 87% of all patients were satisfied with asking about their experiences with medicines and their effects and side effects. Largely to completely satisfied with the received information about SFN was observed in almost 91% of the patients. Almost all patients were satisfied about the skin biopsy taken with the necessary expertise (98%). Nearly 25% of the patients was not or a little satisfied with discussing both the medicines and other

**Table 2. Domain B of the SFN-PSQ: Waiting List Period.**

| | Overall | SFN | No SFN | P |
|---|---|---|---|---|
| | (N = 98) | (n = 82) | (n = 16) | |
| **Acknowledgment with expected time to neurological analysis (n,%)** | | | | 0.818 |
| No, not at all | 6 (6.1) | 5 (6.1) | 1 (6.3) | |
| A little | 5 (5.1) | 4 (4.9) | 1 (6.3) | |
| Largely | 11 (11.2) | 8 (9.8) | 3 (18.8) | |
| Yes, completely | 72 (73.5) | 62 (75.6) | 10 (62.5) | |
| Not applicable | 4 (4.1) | 3 (3.7) | 1 (6.3) | |
| **Satisfied with the information about the neurological analysis (n,%)** | | | | 0.817 |
| No, not at all | 2 (2.0) | 2 (2.4) | 0 (0.0) | |
| A little | 4 (4.1) | 4 (4.9) | 0 (0.0) | |
| Largely | 21 (21.4) | 18 (22.0) | 3 (18.8) | |
| Yes, completely | 62 (63.3) | 51 (62.2) | 11 (68.8) | |
| Not applicable | 9 (9.2) | 7 (8.5) | 2 (12.5) | |
| **4. Searched the internet for information about SFN (n,%)** | | | | 0.335 |
| No, not at all | 4 (4.1) | 3 (3.7) | 1 (6.3) | |
| A little | 22 (22.4) | 16 (19.5) | 6 (37.5) | |
| Largely | 15 (15.3) | 14 (17.1) | 1 (6.3) | |
| Yes, completely | 57 (58.2) | 49 (59.8) | 8 (50.0) | |
| **5. Arrangements, at home or at work, to be able to come to the hospital (n,%)** | | | | 0.712 |
| No, not at all | 22 (22.4) | 20 (24.4) | 2 (12.5) | |
| A little | 16 (16.3) | 12 (14.6) | 4 (25.0) | |
| Largely | 6 (6.1) | 5 (6.1) | 1 (6.3) | |
| Yes, completely | 44 (44.9) | 36 (43.9) | 8 (50.0) | |
| Not applicable | 10 (10.2) | 9 (11.0) | 1 (6.3) | |
| **Which things to arrange (n, %)** | | | | |
| Transport | 11 (34.4) | 11 (13.4) | 0 (0.0) | |
| Work leave | 9 (28.1) | 8 (9.8) | 1 (6.3) | |
| (Hotel) overnight stay | 8 (25.0) | 7 (8.5) | 1 (6.3) | |
| Babysitter | 3 (9.4) | 3 (3.7) | 0 (0.0) | |
| Cancel appointments | 1 (3.1) | 1 (1.2) | 0 (0.0) | |
| **6. Internet link for additional information about SFN (n, %)** | | | | 0.922 |
| No, not at all | 7 (7.1) | 6 (7.3) | 1 (6.3) | |
| A little | 3 (3.1) | 3 (3.7) | 0 (0.0) | |
| Largely | 9 (9.2) | 8 (9.8) | 1 (6.3) | |
| Yes, completely | 74 (75.5) | 61 (74.4) | 13 (81.3) | |
| Not applicable | 4 (5.1) | 4 (4.9) | 1 (6.3) | |
| **7. Waiting time until the neurological analysis acceptable (n, %)** | | | | 0.591 |
| No, not at all | 40 (40.8) | 34 (41.5) | 6 (37.5) | |
| A little | 26 (26.5) | 21 (25.6) | 5 (31.1) | |
| Largely | 14 (14.3) | 13 (15.9) | 1 (6.3) | |
| Yes, completely | 16 (16.3) | 13 (15.9) | 3 (18.8) | |
| Not applicable | 2 (2.0) | 1 (1.2) | 1 (6.3) | |
| **Waiting time in months (mean, SD)** | 10.5 (5.5) | 10.8 (5.5) | 9.4 (5.3) | 0.125 |

SFN: Small Fiber Neuropathy. SD: Standard Deviation.

**Table 3. Domain C of the SFN-PSQ: Reception area and time in the waiting area.**

| | Overall | SFN | No SFN | P |
|---|---|---|---|---|
| | (N = 98) | (n = 82) | (n = 16) | |
| **8. Department or outpatient clinic easily to find in the hospital (n,%)** | | | | **0.005** |
| No, not at all | 3 (3.1) | 3 (3.7) | 0 (0.0) | |
| A little | 14 (14.3) | 8 (9.8) | 6 (37.5) | |
| Largely | 12 (12.2) | 10 (12.2) | 2 (12.5) | |
| Yes, completely | 68 (69.4) | 61 (74.4) | 7 (43.8) | |
| Not applicable | 1 (1.0) | 0 (0.0) | 1 (6.3) | |
| **9. Was the receptionist helpful (n,%)** | | | | 0.068 |
| No, not at all | 3 (3.1) | 1 (1.2) | 0 (0.0) | |
| A little | 14 (14.3) | 0 (0.0) | 0 (0.0) | |
| Largely | 12 (12.2) | 1 (1.2) | 1 (6.3) | |
| Yes, completely | 68 (69.4) | 80 (97.6) | 14 (87.5) | |
| Not applicable | 1 (1.0) | 0 (0.0) | 1 (6.3) | |
| **10. Enough privacy at the front desk (n,%)** | | | | 0.306 |
| No, not at all | 4 (4.1) | 4 (4.9) | 0 (0.0) | |
| A little | 5 (5.1) | 3 (3.7) | 2 (12.5) | |
| Largely | 16 (16.3) | 13 (15.9) | 3 (18.8) | |
| Yes, completely | 71 (72.4) | 61 (74.4) | 10 (62.5) | |
| Not applicable | 2 (2.0) | 1 (1.2) | 1 (6.3) | |
| **11. For day admissions: Satisfied about the welcome of the staff member who attended to you throughout the day (n,%)** | | | | **0.023** |
| A little | 2 (2.0) | 2 (2.4) | 0 (0.0) | |
| Largely | 3 (3.1) | 2 (2.4) | 1 (6.3) | |
| Yes, completely | 75 (76.5) | 67 (81.7) | 8 (50.0) | |
| Not applicable | 18 (18.4) | 11 (13.4) | 7 (43.8) | |
| **12. Staff member skilled in drawing blood (n,%)** | | | | 0.391 |
| A little | 3 (3.1) | 3 (3.7) | 0 (0.0) | |
| Largely | 6 (6.1) | 4 (4.9) | 2 (12.5) | |
| Yes, completely | 89 (90.8) | 75 (91.5) | 14 (87.5) | |
| **13. Blood test inconvenient or unpleasant (n,%)** | | | | 0.641 |
| No, not at all | 76 (77.6) | 65 (79.3) | 11 (68.8) | |
| A little | 20 (20.4) | 15 (18.3) | 5 (31.3) | |
| Largely | 1 (1.0) | 1 (1.2) | 0 (0.0) | |
| Yes, completely | 1 (1.0) | 1 (1.2) | 0 (0.0) | |
| **14. Enough personal attention (n, %)** | | | | 0.337 |
| A little | 4 (4.1) | 4 (4.9) | 0 (0.0) | |
| Largely | 6 (6.1) | 6 (7.3) | 0 (0.0) | |
| Yes, completely | 88 (89.8) | 72 (87.8) | 16 (100.0) | |

*(Continued)*

**Table 3.** (Continued)

| | Overall | SFN | No SFN | P |
|---|---|---|---|---|
| | (N = 98) | (n = 82) | (n = 16) | |
| **15. For day admissions: Satisfied with the information and guidance from the staff member who attended you throughout the day (n, %)** | | | | 0.066 |
| No, not at all | 1 (1.0) | 1 (1.2) | 0 (0.0) | |
| A little | 1 (1.0) | 1 (1.2) | 0 (0.0) | |
| Largely | 4 (4.1) | 4 (4.9) | 0 (0.0) | |
| Yes, completely | 74 (75.5) | 65 (79.3) | 9 (56.3) | |
| Not applicable | 18 (18.4) | 11 (13.4) | 7 (43.8) | |
| **16. Adequate facilities in the waiting room for you and your partner (n,%)** | | | | 0.943 |
| No, not at all | 1 (1.0) | 1 (1.2) | 0 (0.0) | |
| A little | 6 (6.1) | 5 (6.1) | 1 (6.3) | |
| Largely | 13 (13.3) | 10 (12.2) | 3 (18.8) | |
| Yes, completely | 73 (74.5) | 62 (75.6) | 11 (68.8) | |
| Not applicable | 5 (5.1) | 4 (4.9) | 1 (6.3) | |
| **Explanation: Which comments (n,%)** | | | | |
| Too little services (Covid-19 period) | 13 (61.9) | 11 (13.4) | 2 (12.5) | |
| Difficulties with blood draw | 3 (14.3) | 2 (2.4) | 1 (6.3) | |
| No timeframe for investigations | 2 (9.5) | 2 (2.4) | 0 (0.0) | |
| No or unprofessional staff | 3 (14.3) | 3 (3.7) | 0 (0.0) | |

SFN-PSQ: Small Fiber Neuropathy Patient Satisfaction Questionnaire.

treatments. A majority of all patients (ca. 84%) missed no other aspects of the treatment of your symptoms.

## G-H. Separate Consultation with the Doctor or NP about Psychological Symptoms

A total of 19 patients, with a majority of patients with SFN (ca. 90%), had a separate consultation with the Doctor or NP about psychological symptoms, see Table 6. Almost 90% of the patients was completely satisfied with the personal attention, expertise and privacy during the consultation. The reason for the consultation was in nearly 90% of the patients largely to completely clear. In more than 73% of the patients the received information was largely to completely tailored to their personal situation and in 63% of the patients the consultation had added value.

## I. Going Home

More than 90% of all patients went home largely to completely satisfied, and almost 85% knew whom to contact with questions or problems (Table 7). A majority of all patients (96%) received the results of the neurological analysis by telephone consultation, and almost 94% was largely to completely satisfied about discussing the results by phone. Nearly 45% of all patients was not or a little satisfied about the waiting period of 7–8 weeks for receiving the results, and 98% of all patients was largely to completely satisfied about the GP, the neurologist and

**Table 4. Domain D and E of the SFN-PSQ: Chest X-ray and nerve tests.**

| | Overall (N = 98) | SFN (n = 82) | No SFN (n = 16) | P |
|---|---|---|---|---|
| **DOMAIN D: Chest X-ray** | | | | |
| **17. Satisfied with the welcome by the staff member (n,%)** | | | | 0.626 |
| No, not at all | 1 (1.0) | 1 (1.2) | 0 (0.0) | |
| A little | 2 (2.0) | 2 (2.4) | 0 (0.0) | |
| Largely | 5 (5.1) | 5 (6.1) | 0 (0.0) | |
| Yes, completely | 78 (79.6) | 63 (76.8) | 15 (93.8) | |
| Not applicable | 12 (12.2) | 11 (13.4) | 1 (6.3) | |
| **18. Satisfied with the information and guidance provided by the staff member (n,%)** | | | | 0.712 |
| No, not at all | 1 (1.0) | 1 (1.2) | 0 (0.0) | |
| A little | 2 (2.0) | 2 (2.4) | 0 (0.0) | |
| Largely | 4 (4.1) | 4 (4.1) | 0 (0.0) | |
| Yes, completely | 80 (81.6) | 65 (79.3) | 15 (93.8) | |
| Not applicable | 11 (11.2) | 10 (12.2) | 1 (6.3) | |
| **19. Enough personal attention (n,%)** | | | | 0.507 |
| No, not at all | 1 (1.0) | 1 (1.2) | 0 (0.0) | |
| A little | 6 (6.1) | 6 (7.3) | 0 (0.0) | |
| Largely | 6 (6.1) | 4 (4.9) | 2 (12.5) | |
| Yes, completely | 73 (74.5) | 60 (73.2) | 13 (81.3) | |
| Not applicable | 12 (12.2) | 11 (13.4) | 1 (6.3) | |
| **20. Staff member performing the chest x-ray had the necessary expertise (n,%)** | | | | 0.515 |
| A little | 2 (2.0) | 2 (2.4) | 0 (0.0) | |
| Largely | 4 (4.1) | 4 (4.9) | 0 (0.0) | |
| Yes, completely | 79 (80.6) | 64 (78.0) | 15 (93.8) | |
| Not applicable | 13 (13.3) | 12 (14.6) | 1 (6.3) | |
| **DOMAIN E: Nerve Tests (Nerve Conduction Studies)** | | | | |
| **21. Satisfied with the welcome by the staff member (n,%)** | | | | 0.055 |
| A little | 2 (2.0) | 1 (1.2) | 1 (6.3) | |
| Largely | 8 (8.2) | 7 (8.5) | 1 (6.3) | |
| Yes, completely | 85 (86.7) | 73 (89.0) | 12 (75.0) | |
| Not applicable | 3 (3.1) | 1 (1.2) | 2 (12.5) | |
| **22. Satisfied with the information and guidance of the staff member (n,%)** | | | | 0.086 |
| A little | 3 (3.1) | 2 (2.4) | 1 (6.3) | |
| Largely | 9 (9.2) | 8 (9.8) | 1 (6.3) | |
| Yes, completely | 83 (84.7) | 71 (86.6) | 12 (75.0) | |
| Not applicable | 3 (3.1) | 1 (1.2) | 2 (12.5) | |
| **23. Enough personal attention (n, %)** | | | | 0.072 |
| A little | 4 (4.1) | 4 (4.9) | 0 (0.0) | |
| Largely | 8 (8.2) | 6 (7.3) | 2 (12.5) | |
| Yes, completely | 83 (84.7) | 71 (86.6) | 12 (75.0) | |
| Not applicable | 3 (3.1) | 1 (1.2) | 2 (12.5) | |
| **24. Doctor performing the test(s) had the necessary expertise (n,%)** | | | | **0.016** |
| A little | 2 (2.0) | 2 (2.4) | 0 (0.0) | |
| Largely | 9 (9.2) | 6 (7.3) | 3 (18.8) | |
| Yes, completely | 82 (83.7) | 72 (87.8) | 10 (62.5) | |
| Not applicable | 5 (5.1) | 2 (2.4) | 3 (18.8) | |

SFN-PSQ: Small Fiber Neuropathy Patient Satisfaction Questionnaire.

NCS: Nerve Conduction Studies. QST: Quantative Sensory Testing.

**Table 5. Domain F of the SFN-PSQ: Consultation with the Doctor or NP.**

| | Overall | SFN | No SFN | P |
|---|---|---|---|---|
| | (N = 98) | (n = 82) | (n = 16) | |
| **25. Enough personal attention (n,%)** | | | | 0.666 |
| A little | 2 (2.0) | 2 (2.4) | 0 (0.0) | |
| Largely | 2 (2.0) | 2 (2.4) | 0 (0.0) | |
| Yes, completely | 94 (96.0) | 78 (95.1) | 16 (100.0) | |
| **26. Enough expertise of the Doctor or NP (n,%)** | | | | 0.726 |
| A little | 3 (3.1) | 3 (3.7) | 0 (0.0) | |
| Largely | 7 (7.1) | 6 (7.3) | 1 (6.3) | |
| Yes, completely | 88 (89.8) | 73 (98.0) | 15 (93.8) | |
| **27. Enough privacy during the consultation (n,%)** | | | | 0.528 |
| Largely | 2 (2.0) | 2 (2.4) | 0 (0.0) | |
| Yes, completely | 96 (98.0) | 80 (97.6) | 16 (100.0) | |
| **28. Asked about your own ideas, expectations or experiences (n,%)** | | | | 0.873 |
| No, not at all | 2 (2.0) | 2 (2.4) | 0 (0.0) | |
| A little | 1 (1.0) | 1 (1.2) | 0 (0.0) | |
| Largely | 17 (17.4) | 15 (18.3) | 2 (12.5) | |
| Yes, completely | 77 (78.6) | 63 (76.8) | 14 (87.5) | |
| Not applicable | 1 (1.0) | 1 (1.2) | 0 (0.0) | |
| **29. Asked about the impact of your symptoms on your daily life (n,%)** | | | | 0.846 |
| No, not at all | 1 (1.0) | 1 (1.2) | 0 (0.0) | |
| A little | 3 (3.1) | 3 (3.7) | 0 (0.0) | |
| Largely | 18 (18.4) | 15 (18.3) | 3 (18.8) | |
| Yes, completely | 76 (77.6) | 63 (76.8) | 13 (81.3) | |
| **30. Asked about your lifestyle (including smoking, exercise, diet) (n,%)** | | | | 0.740 |
| No, not at all | 2 (2.0) | 2 (2.4) | 0 (0.0) | |
| A little | 2 (2.0) | 2 (2.4) | 0 (0.0) | |
| Largely | 18 (18.4) | 14 (17.1) | 4 (25.0) | |
| Yes, completely | 76 (77.6) | 64 (78.0) | 12 (75.0) | |
| **31. Asked about your experiences with medicines, their effects or side effects (n,%)** | | | | 0.777 |
| No, not at all | 4 (4.1) | 4 (4.9) | 0 (0.0) | |
| A little | 4 (4.1) | 4 (4.9) | 0 (0.0) | |
| Largely | 13 (13.3) | 11 (13.4) | 2 (12.5) | |
| Yes, completely | 72 (73.5) | 59 (72.0) | 13 (81.3) | |
| Not applicable | 5 (5.1) | 4 (4.9) | 1 (6.3) | |
| **32. Satisfied with the information you received about SFN (n,%)** | | | | 0.584 |
| No, not at all | 2 (2.0) | 2 (2.4) | 0 (0.0) | |
| A little | 6 (6.1) | 6 (7.3) | 0 (0.0) | |
| Largely | 25 (25.5) | 22 (26.8) | 3 (18.8) | |
| Yes, completely | 64 (65.3) | 51 (62.2) | 13 (81.3) | |
| Not applicable | 1 (1.0) | 1 (1.2) | 0 (0.0) | |
| **Explanation: Which comments (n,%)** | | | | |
| Information about SFN did not matched expectations | 6 (6.1) | 6 (7.3) | 0 (0.0) | |
| No clear answer to what SFN means, find out for yourself | 5 (5.1) | 3 (3.7) | 2 (12.5) | |
| Only symptomatic treatment | 1 (1.0) | 1 (1.2) | 0 (0.0) | |

*(Continued)*

**Table 5.** (Continued)

|  | Overall | SFN | No SFN | P |
|---|---|---|---|---|
|  | (N = 98) | (n = 82) | (n = 16) |  |
| I nsufficient attention to certain topics (including personal experiences, causes, medication) | 3 (3.1) | 3 (3.7) | 0 (0.0) |  |
| Employee had a bad day | 1 (1.0) | 1 (1.2) | 0 (0.0) |  |
| EMG was painful | 1 (1.0) | 1 (1.2) | 0 (0.0) |  |
| No SFN specialist (substitute) | 2 (2.0) | 2 (2.4) | 0 (0.0) |  |
| **33. Skin biopsy taken with the necessary expertise (n,%)** |  |  |  | 0.528 |
| Largely | 2 (2.0) | 2 (2.4) | 0 (0.0) |  |
| Yes, completely | 96 (98.0) | 80 (97.6) | 16 (100.0) |  |
| **34. Medicines and other treatments both discussed with you (n,%)** |  |  |  | 0.221 |
| No, not at all | 10 (10.2) | 7 (8.5) | 3 (18.8) |  |
| A little | 14 (14.3) | 13 (15.9) | 1 (6.3) |  |
| Largely | 14 (14.3) | 14 (17.1) | 0 (0.0) |  |
| Yes, completely | 52 (53.1) | 72 (51.2) | 10 (62.5) |  |
| Not applicable | 8 (8.2) | 6 (7.3) | 2 (12.5) |  |
| **35. Missed other aspects of the treatment of your symptoms (n,%)** |  |  |  | 0.549 |
| No, not at all | 63 (64.3) | 53 (64.4) | 10 (62.5) |  |
| A little | 19 (19.4) | 15 (18.3) | 4 (25.0) |  |
| Largely | 4 (4.1) | 4 (4.9) | 0 (0.0) |  |
| Yes, completely | 2 (2.0) | 1 (1.2) | 1 (6.3) |  |
| Not applicable | 10 (10.2) | 9 (11.0) | 1 (6.3) |  |

SFN-PSQ: Small Fiber Neuropathy Patient Satisfaction Questionnaire. EMG: Electromyography.

themselves receiving the same letter with results. Nearly 82% largely to completely satisfied with the advice and/or information on further treatment.

## J. General

Almost 85% of all patients was mostly to completely satisfied with the neurological analysis as a whole, with an overall patient satisfaction score of 8.7 (IQR 8–10) (from 0 till 10 maximum) (Table 8). Nearly 90% of all patients were willing to participate again in scientific research on SFN. One third of the patients (n = 30) gave additional comments, which involved (dis)satisfaction about various reasons: diagnosis of polyneuropathy instead of SFN, QST done in another hospital, NPs instead of neurologists, waiting time between investigations too long, bad lunch, no specific diagnostic examinations adapted to complaints, no SFN-diagnose (n = 10); general dissatisfaction about waiting times (n = 7); overall patient satisfaction with neurological analysis (n = 6); willingness to participate in scientific research (n = 4); and lack of knowledge of SFN among doctors (n = 3). In total, one fifth (n = 20) gave negative feedback about the waiting list and the waiting time between studies at the time of the neurological analysis.

## Statistically significant differences with patient satisfaction

There were no statistically significant differences found in all patients in the domains Waiting List Period, Consultation with the Doctor/ NP, Chest X-ray, Consultation with the Doctor/ NP, Consultation with the Doctor or NP about Psychological Symptoms, General, and patient

**Table 6. Domain G and H of the SFN-PSQ: Separate Consultation with the Doctor or NP about Psychological Symptoms.**

| | Overall | SFN | No SFN | P |
|---|---|---|---|---|
| | (N = 19) | (n = 82) | (n = 16) | |
| **36. Enough personal attention (n,%)** | | | | 0.200 |
| Yes, completely | 17 (89.5) | 16 (19.5) | 1 (6.3) | |
| Not applicable | 2 (10.5) | 66 (80.5) | 15 (93.8) | |
| **37. Enough privacy during the consultation (n,%)** | | | | 0.200 |
| Yes, completely | 17 (89.5) | 16 (19.5) | 1 (6.3) | |
| Not applicable | 2 (10.5) | 66 (80.5) | 15 (93.8) | |
| **38. Doctor or NP had the necessary expertise (n,%)** | | | | 0.200 |
| Yes, completely | 17 (89.5) | 16 (19.5) | 1 (6.3) | |
| Not applicable | 2 (10.5) | 66 (80.5) | 15 (93.8) | |
| **39. Reason for the consultation was clear to you (n,%)** | | | | 0.086 |
| No, not at all | 1 (5.3) | 1 (1.2) | 0 (0.0) | |
| Largely | 1 (5.3) | 0 (0.0) | 1 (6.3) | |
| Yes, completely | 16 (84.2) | 15 (18.3) | 1 (6.3) | |
| Not applicable | 1 (5.3) | 66 (80.5) | 14 (87.5) | |
| **40. Information received was tailored to your personal situation (n,%)** | | | | 0.097 |
| Largely | 5 (5.1) | 3 (3.7) | 2 (12.5) | |
| Yes, completely | 13 (68.4) | 13 (15.9) | 0 (0.0) | |
| Not applicable | 1 (5.3) | 66 (80.5) | 14 (87.5) | |
| **41. Did the consultation had added value to you (n,%)** | | | | 0.294 |
| A little | 5 (26.3) | 4 (4.9) | 1 (6.3) | |
| Largely | 2 (10.5) | 1 (1.2) | 1 (6.3) | |
| Yes, completely | 10 (52.6) | 10 (12.2) | 0 (0.0) | |
| Not applicable | 2 (10.5) | 69 (81.7) | 14 (87.5) | |

SFN-PSQ: Small Fiber Neuropathy Patient Satisfaction Questionnaire.

satisfaction. Statistically significant less satisfaction among the patients with no SFN concerned the accessibility to the NDCU or OC, the staff member who attended them throughout the day at the NDCU, and the expertise of the doctor who performed the NCS/ QST (all p < 0.05). Patients with SFN were statistically significant less satisfied about the waiting period of 7–8 weeks for receiving the results (p < 0.05). There were no statistically significant associations found between age, sex, education level, anxiety and depression scores, and overall patient satisfaction. A statistically significant association was found between the general health status and overall patient satisfaction (p < 0.05).

## Discussion

This is the first study examining the patient satisfaction and patient accessibility of a targeted diagnostic SFN service in detail, measured by a new feasible and valid SFN patient satisfaction questionnaire, the SFN-PSQ.

Results showed that the overall patient satisfaction of the diagnostic SFN service was high and is of an outstanding quality from patient's perspective. Especially the domains (B) 'Waiting List Period', (D) Chest X-ray', (F) 'Consultation with the Doctor or NP', (G-H) 'Separate Consultation with the Doctor or NP about Psychological Symptoms', and (J) 'General' had a high patient satisfaction. This might be explained by the essential role of the provider attitude, technical competence, and efficacy of the SFN diagnostic service, which are considered of high

**Table 7. Domain I of the SFN-PSQ: Going Home.**

| | Overall | SFN | No SFN | P |
|---|---|---|---|---|
| | (N = 98) | (n = 82) | (n = 16) | |
| **42. Satisfied when going home (n,%)** | | | | 0.910 |
| No, not at all | 1 (1.0) | 1 (1.2) | 0 (0.0) | |
| A little | 6 (6.1) | 5 (6.1) | 1 (6.3) | |
| Largely | 17 (17.3) | 15 (18.3) | 2 (12.5) | |
| Yes, completely | 74 (75.5) | 61 (74.4) | 13 (81.3) | |
| **43. Know whom to contact with questions or problems (n,%)** | | | | 0.192 |
| No, not at all | 7 (7.1) | 7 (8.5) | 0 (0.0) | |
| A little | 6 (6.1) | 6 (7.3) | 0 (0.0) | |
| Largely | 9 (9.2) | 6 (7.3) | 3 (18.8) | |
| Yes, completely | 74 (75.5) | 62 (75.6) | 12 (75.0) | |
| Not applicable | 2 (2.0) | 1 (1.2) | 1 (6.3) | |
| **44. How did you received the results of the SFN analysis (n,%)** | | | | 0.528 |
| Telephone consultation | 96 (98.0) | 80 (97.6) | 16 (100.0) | |
| Outpatient consultation | 2 (2.0) | 2 (2.4) | 0 (0.0) | |
| **45. Satisfied with how the results were discussed (n,%)** | | | | 0.919 |
| No, not at all | 1 (1.0) | 1 (1.2) | 0 (0.0) | |
| A little | 2 (2.0) | 4 (4.9) | 1 (6.3) | |
| Largely | 20 (20.4) | 16 (19.5) | 4 (25.0) | |
| Yes, completely | 72 (73.5) | 61 (74.4) | 11 (68.8) | |
| **46. Do you find the 8-week period for receiving the results acceptable (n,%)** | | | | **0.016** |
| No, not at all | 11 (11.2) | 8 (9.8) | 3 (18.8) | |
| A little | 33 (33.7) | 33 (40.2) | 0 (0.0) | |
| Largely | 29 (29.6) | 23 (28.0) | 6 (37.5) | |
| Yes, completely | 25 (25.5) | 18 (22.0) | 7 (43.8) | |
| **47. Do you think it is enough if the GP and the neurologist receive the same letter with results as you (n,%)** | | | | 0.932 |
| No, not at all | 1 (1.0) | 1 (1.2) | 0 (0.0) | |
| A little | 1 (1.0) | 1 (1.2) | 0 (0.0) | |
| Largely | 5 (5.1) | 4 (4.9) | 1 (6.3) | |
| Yes, completely | 91 (92.9) | 76 (92.7) | 15 (93.8) | |
| **48. Satisfied with the advice/information on further treatment (n,%)** | | | | 0.280 |
| No, not at all | 3 (3.1) | 3 (3.7) | 0 (0.0) | |
| A little | 11 (11.2) | 8 (9.8) | 3 (18.8) | |
| Largely | 17 (17.3) | 12 (14.6) | 5 (31.3) | |
| Yes, completely | 63 (64.3) | 55 (67.1) | 8 (50.0) | |
| Not applicable | 4 (4.1) | 4 (4.9) | 0 (0.0) | |

SFN-PSQ: Small Fiber Neuropathy Patient Satisfaction Questionnaire. GP: General Practitioner.

importance in patient satisfaction [30]. Overall, the general health status was statistically significant associated with the patient satisfaction of the diagnostic SFN service.

Patients with SFN were statistically significant less satisfied about an item of the domain 'Going Home' about the 8-week period for receiving the results. This waiting time for discussing the results combined with the time on the waiting list means that patients have to wait almost a year to confirm or dismiss the diagnosis of SFN. For other conditions, delayed diagnosis is known to further delay patients' access to appropriate care and effective treatments for the condition [31–33], which could also apply to SFN. Desirable is to speed up the 8-week period, but this would only be possible if genetic testing and pathology results are known earlier.

**Table 8. Domain J of the SFN-PSQ: General.**

| | Overall | SFN | No SFN | P |
|---|---|---|---|---|
| | (N = 98) | (n = 82) | (n = 16) | |
| 49. Indicate your satisfaction with the neurological analysis as a whole (n,%) | | | | 0.965 |
| No, not at all (1–3,5) | 1 (1.0) | 1 (1.2) | 0 (0.0) | |
| A little bit (3,5–6) | 5 (5.1) | 4 (4.9) | 1 (6.3) | |
| Mostly (6–8,5) | 22 (22.4) | 20 (24.4) | 4 (18.8) | |
| Yes, completely (8,5–10) | 61 (62.2) | 51 (62.2) | 10 (62.5) | |
| Missing | 9 (9.2) | 8 (9.8) | 1 (6.3) | |
| **Mean score (IQR)** | 8.7 (8–10) | 8.7 (8–10) | 8.7 (8–10) | |
| 50. Willing to participate again in scientific research on SFN (n,%) | | | | 0.217 |
| Yes | 88 (89.8) | 75 (91.5) | 13 (81.3) | |

SFN-PSQ: Small Fiber Neuropathy Patient Satisfaction Questionnaire. IQR: Inter Quartile Range.

The accessibility to the NDCU or OC, the attending staff member at the NDCU and the expertise of the doctor who performed the NCS/ QST were for patients with no SFN statistically significant less satisfying. A poor self-rated health status could be a possible clarification, as this is associated with a worse patient experience [34–36], which was applicable in more than half of the patients with no SFN and also statistically significant associated with the patient satisfaction in our study.

The item on waiting time of the domain 'Waiting List Period' scored less on patient satisfaction in all patients visiting the NDCU, probably due to a waiting time of more than ten months. Since SFN-complaints are non-specific and may vary substantially, patients often visit several caregivers before they are referred to the only specialized SFN center in the Netherlands with a limited access. These findings are in line with a Canadian study about access to multidisciplinary pain treatment facilities, and their results indicated that longer duration of pain, lower family income and referral from the GP were significantly associated with longer waiting times [37]. However, these variables were not included in the current study and, as our SFN Center is a tertiary referral center, patients are only referred by neurologists from other hospitals and therefore these findings are not entirely comparable.

Another study concluded that a gap of six months or more before getting neuropathic pain treatment was associated with a worsening of health-related quality of life and psychological symptoms [38]. For a more efficient organization of the SFN diagnostic service, an administrative improvement has been made by digitizing the triages. Furthermore, a solution for reducing the long waiting list could be a combination of investment and efficient use of existing resources, such as e-health and self-management in teleneurology, for increasing the diagnostic capacities for SFN [39, 40].

A global multicenter survey of patients with migraine suggested possible solutions to long waiting lists, such as appropriate patient referral, triage and priority allocation, nurse role and task delegation, and follow-up management [41]. The majority of all referred patients with SFN-related complaints to the SFN Center in the Netherlands are triaged by and assigned to NPs. The unique NP competences of assessing referrals, clinically diagnosing SFN, managing tasks for the continuity of care, and contributing to various chronic pain management options, can contribute to a higher quality of care and increase patient satisfaction [42, 43].

A quarter of all patients were less satisfied with an item of the domain 'Consultation with the Doctor or NP', namely discussing both medicines and other treatments. One of the explanations could be that some patients do not want to take neuropathic pain medication because

it consists of antidepressants and anticonvulsants [6, 7]. The adherence to these drugs is sub-optimal with approximately 40% medication adherence [44], which may be due to multiple factors such as treatment complexity, co-payment, effectiveness and side effects of the medication, and inadequate follow-up [45]. In these cases, patients may not want to be informed about the possible treatments and probably do not ask any questions about this during the consultation with the Doctor or NP.

## Strength and limitations

A strength of this study is that the results are collected with a feasible and reliable instrument on patient satisfaction in patients with SFN-related symptoms. There was a high response rate in a large patient group and the time of filling out the SFN-PSQ was within the recommendation of < 30 minutes for keeping an intact interest and attention of the participant [46]. Therefore the results provide a good reflection of the patient satisfaction of the SFN diagnostic service. In addition, important points for improvement have been extracted from the results of the online survey, allowing the SFN diagnostic service to be optimized in the future.

In an effort to shorten the waiting list for the diagnostic SFN service, the waiting list is checked more often for the accessibility of patients and the presence of their SFN-related complaints. In addition, the triages have been tightened up by checking whether all the required data is present. Finally, additional neurological analyses are planned when more staff is available, for example in the presence of a fellow neurologist.

A limitation of this study is that it concerns a newly developed patient satisfaction questionnaire, with known deficiencies of ordinal data [47–50]. In view of an SFN-PSQ with a one-dimensional metric, free from item bias and without unordered thresholds or local dependence, it is recommended that the SFN-PSQ via Rasch modeling will be analyzed in a future study.

Also this study consists of a single center study, but this could not have been carried out otherwise, since our center is the only SFN expertise center in the Netherlands to date. Conversely, the results are a good reflection of patients' experiences with the diagnostic SFN service.

For future use of the SFN-PSQ, its logistic and diagnostic pathway of the neurological analysis of SFN could be an example for other hospitals for improving the quality of care, continuity and access to SFN health care. However, it is important to find out in future research whether the SFN diagnostic service also meets the needs and expectations of patients from a cross-cultural setting.

Furthermore, access to neurological care is already scarce, and its supply and distribution are unlikely to match its future demand [39]. The results of this study show that patients prefer a shorter waiting time for a neurological SFN analysis, and that additional investigations can be performed in their own hospital. By adapting or aligning the logistical and diagnostic processes of the neurological SFN analysis in other hospitals with the SFN diagnostic service, there would be many benefits, such as reducing waiting time and travel distance, and nevertheless spreading the knowledge and expertise of SFN. In addition, capacity may be enhanced through a combination of investment and more efficient use of existing resource, such as digital health care (eHealth) and self-management, such as teleneurology [39]. Randomized controlled trials of teleneurology have demonstrated that teleneurology is feasible, generates outcomes comparable to usual care and is well liked by patients and clinicians [51, 52]. Future research should aim to use patient satisfaction questionnaires, such as the SFN-PSQ, to optimize the logistic and diagnostic pathway of the neurological analysis of SFN, which might help hospitals to improve their quality of SFN care.

## Supporting information

**S1 Appendix.**
(PDF)

## Acknowledgments

The authors thank Carla M.L. Gorissen-Brouwers and Aysun Damci for their expert substantive contribution and for recruiting the patients for this study.

## Author Contributions

**Conceptualization:** Margot Geerts, Janneke G. J. Hoeijmakers, Brigitte A. B. Essers, Ingemar S. J. Merkies, Catharina G. Faber, Mariëlle E. J. B. Goossens.

**Data curation:** Margot Geerts, Mariëlle E. J. B. Goossens.

**Formal analysis:** Margot Geerts, Janneke G. J. Hoeijmakers, Brigitte A. B. Essers, Ingemar S. J. Merkies, Catharina G. Faber, Mariëlle E. J. B. Goossens.

**Investigation:** Margot Geerts, Ingemar S. J. Merkies, Catharina G. Faber, Mariëlle E. J. B. Goossens.

**Methodology:** Margot Geerts, Janneke G. J. Hoeijmakers, Brigitte A. B. Essers, Ingemar S. J. Merkies, Catharina G. Faber, Mariëlle E. J. B. Goossens.

**Resources:** Janneke G. J. Hoeijmakers, Brigitte A. B. Essers, Catharina G. Faber, Mariëlle E. J. B. Goossens.

**Software:** Margot Geerts, Catharina G. Faber, Mariëlle E. J. B. Goossens.

**Supervision:** Janneke G. J. Hoeijmakers, Brigitte A. B. Essers, Ingemar S. J. Merkies, Catharina G. Faber, Mariëlle E. J. B. Goossens.

**Validation:** Brigitte A. B. Essers, Ingemar S. J. Merkies, Mariëlle E. J. B. Goossens.

**Writing – original draft:** Margot Geerts, Brigitte A. B. Essers, Ingemar S. J. Merkies, Catharina G. Faber, Mariëlle E. J. B. Goossens.

**Writing – review & editing:** Janneke G. J. Hoeijmakers, Brigitte A. B. Essers, Ingemar S. J. Merkies, Catharina G. Faber, Mariëlle E. J. B. Goossens.

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
