## [Decision Letter · Decision Letter 0]

1 Feb 2024

Patient satisfaction and patient accessibility in a small fiber neuropathy diagnostic service in the Netherlands: a single-center, prospective, survey-based cohort study

PONE-D-23-42238

Dear Dr. Geerts,

We’re pleased to inform you that your manuscript has been judged scientifically suitable for publication and will be formally accepted for publication once it meets all outstanding technical requirements.

Kind regards,

Dr. Vara Prasad Saka

Academic Editor

PLOS ONE

1. Thank you for stating the following in the Competing Interests section:

“I have read the journal's policy and the authors of this manuscript have the following competing interests: Janneke G.J. Hoeijmakers reports a grant from the Prinses Beatrix Spierfonds (W.OK17-09), outside the submitted work.

Catharina G. Faber reports grants from European Union’s Horizon 2020 research and innovation program Marie Sklodowska-Curie grant for PAIN-Net, Molecule-to-man pain network (grant no. 721841), grants from Prinses Beatrix Spierfonds, grants from Grifols and Lamepro for a trial on IVIg in small fiber neuropathy, other from Steering committees/advisory board for studies in small fiber neuropathy of Biogen/Convergence and Vertex, outside the submitted work.

Ingemar S.J. Merkies reports grants and non-financial support from Grifols, grants from Lamepro, during the conduct of the study; other from Participation in steering committees of the Talecris ICE Study, CSL Behring, LFB, Novartis, Octapharma, Biotest and UCB, outside the submitted work.”

Additional Editor Comments:

Thank you for submitting your manuscript with Manuscript ID PONE-D-23-42238 to PLOS ONE. After a thorough review by the assigned reviewers and careful consideration, I am pleased to inform you that your manuscript has been accepted for publication.

I appreciate the quality of your work and believe it will contribute positively to PLOS ONE.

Congratulations on your accepted manuscript, and I look forward to seeing it published.

Reviewers' comments:

Reviewer's Responses to Questions

**Comments to the Author**

1. Is the manuscript technically sound, and do the data support the conclusions?

Reviewer #1: Yes

Reviewer #2: Yes

2. Has the statistical analysis been performed appropriately and rigorously? 

Reviewer #1: Yes

Reviewer #2: Yes

3. Have the authors made all data underlying the findings in their manuscript fully available?

Reviewer #1: Yes

Reviewer #2: Yes

4. Is the manuscript presented in an intelligible fashion and written in standard English?

Reviewer #1: Yes

Reviewer #2: Yes

5. Review Comments to the Author

Reviewer #1: the research work carried out authors by was technically sound good, used proper statistical methods which are suitable for this type of study design. The Way authors presented the work was satisfied.

Manuscript can be accepted without any revisions

Reviewer #2: I would like to appreciate the authors attempt to provide a detailed report on SFN and Patient satisfaction. Results indicated that overall a significant improvement in health status. But i have one question that why the authors couldn't addressed the cost effective analysis on SFN? Because these conditions are chronic and would impact the socioeconomic status of the family.

6. PLOS authors have the option to publish the peer review history of their article (what does this mean?). If published, this will include your full peer review and any attached files.

Reviewer #1: No

Reviewer #2: **Yes: **Phani Kumar Kola

---

## [Editor Report · Acceptance letter]

13 Feb 2024

PONE-D-23-42238 

PLOS ONE

Dear Dr. Geerts, 

I'm pleased to inform you that your manuscript has been deemed suitable for publication in PLOS ONE. Congratulations! Your manuscript is now being handed over to our production team.

Kind regards, 

on behalf of

Dr. Vara Prasad Saka 

Academic Editor

PLOS ONE